REGISTERED REPORT PROTOCOL

# Predictive variables of prescription opioid misuse in patients with chronic noncancer pain. Development of a risk detection scale: A registered report protocol

**Carmen Ramírez-Maestre**[1,2¤]*, **Alicia E. López-Martínez**[1,2], **Rosa Esteve**[1,2]

**1** Universidad de Málaga, Facultad de Psicología, Andalucía Tech, Málaga, Spain, **2** Instituto de Investigación Biomédica de Málaga, Málaga, Spain

¤ Current address: Facultad de Psicología y Logopedia, Málaga, Spain
* cramirez@uma.es

## Abstract

### Background

Opioid therapy is utilized for a broad range of chronic pain conditions. Several studies have highlighted the adverse effects of opioid medication due to the misuse of these drugs. The gradual increase in the use of opioids has become a global phenomenon and is generating social concern. Several treatment guidelines have strongly recommended assessing the risks and benefits of pharmacological treatment with opioids. These guidelines also recommend the psychological assessment of patients with chronic noncancer pain in order to make informed decisions on the advisability of intervention with opioids. Some authors have emphasized the relevance of assessing the risk of opioid misuse in patients with noncancer chronic pain before initiating treatment.

### Methods and analysis

Two studies will be conducted, each with a different primary objective. The primary objective of the first study (Study 1) will be to conduct a comprehensive investigation to identify the factors most closely associated with subsequent opioid misuse; and based on the results of Study 1, the primary objective of the second study (Study 2) will be to develop a brief, reliable, valid, and useful instrument that would enable health care providers to make decisions on opioid prescription and on the required level of monitoring and follow-up. These decisions would have positive consequences for patient wellbeing. Study 1 will include a logistic regression analysis to test the hypothetical model. Study 2 will have a longitudinal design and include three assessment sessions in order to develop a measure to assess the risk of prescribed opioid misuse and to analyse its reliability and validity. Participants will be individuals with chronic noncancer pain attending three Pain Units. These individuals will either be undergoing pharmacological treatment that includes opioid analgesics (Study 1, N = 400) or are going to commence such treatment (Study 2, N = 250).

**Data Availability Statement:** All relevant data from this study will be made available upon study completion.

**Funding:** This study was supported by grants from the Spanish Ministry of Science and Innovation (PID2019-106086RB-I00), and the Regional Government of Andalusia (HUM-566).

**Competing interests:** No authors have competing interests.

# Introduction

Current empirical evidence shows that chronic pain is a major health issue and one of the most common reasons for patients to seek medical help [1–3]. Chronic pain is pain that lasts or recurs for longer than 3 months [3]. Given the complex nature of chronic pain, its appropriate treatment often requires individualized assessments by multidisciplinary teams capable of addressing the physical and psychological aspects of pain [4, 5]. It also requires pharmacological treatment, which is a common component of chronic pain management regimens [6, 7]. Such treatment often includes the use of nonopioid and opioid analgesics, as well as adjuvant medications used to prevent or treat the adverse effects of analgesics or enhance the effect of opioid analgesics. Opioid therapy is now utilized for a broad range of chronic pain conditions [8]. Several studies have highlighted the adverse effects of opioid medication due to the misuse of these drugs [9–13], which has been defined as the use of opioids in a manner other than how they are prescribed [14]. Prescription opioid misuse may involve a wide range of behaviours, including overuse [14]. In our project, misuse is defined as the overuse of prescribed opioids (i.e., taking more opioids than prescribed). In Spain, there is growing concern about the use and misuse of opioid analgesics [15, 16]. In 2019, the Spanish Ministry of Health Care published a report on medication use. The report analysed opioid consumption based on prescribing data obtained by the Spanish National Health System from private and public health care sources, including hospitals. The results showed an increase in opioid consumption in Spain from 10.02 daily doses/1000 inhabitants/d in 2010 to 18.73 daily doses/1000 inhabitants/d in 2018. This gradual increase in the use of opioids has become a global phenomenon and is generating social concern. Ray et al., [17] found that the increased use of opioids for long-term treatment increases the risk of death due to both unintentional overdose and cardiorespiratory problems. Apart from their conventional use, an even more worrisome issue is prescription opioid misuse. Therefore, prior risk assessment has emerged as a crucial step in making informed decisions on the risks of starting treatment with opioid drugs.

Several factors have been identified as being associated with the increased risk of addictive substance misuse. One of the most widely studied risk factors is a previous history of addiction [12, 18–20]. However, this risk factor alone is unable to completely account for this phenomenon. Other studies have addressed predictors of misuse in relation to age [21, 22], smoking [23], obesity [9], psychological disorders such as depression, anxiety, and Posttraumatic Stress Disorder [24, 25], or a history of childhood abuse [18].

Regarding the misuse of prescribed medications, several studies have shown the predictive value of patients' concerns about medication [26–28]. Higher levels of catastrophizing and the nonacceptance of pain make patients more vulnerable to prescription opioid misuse [29–31]. Empirical research has indicated that anxiety sensitivity is associated with catastrophizing in pain patients [32, 33] and that anxiety sensitivity is a risk factor for substance misuse [34–36]. There is also extensive empirical evidence in support of an association between impulsivity and addiction [37, 38]. Several studies have found a link between impulsivity and opioid misuse [39, 40]. Finally, several studies have analysed sex differences in the use and misuse of opioids [41, 42] and in the correlates of risk for opioid misuse among chronic pain patients prescribed opioids for chronic pain [43, 44].

Some authors have emphasized the relevance of assessing the risk of opioid misuse in patients with noncancer chronic pain before initiating treatment [9, 45]. In fact, several treatment guidelines have strongly recommended assessing the risks and benefits of pharmacological treatment with opioids [9]. These guidelines also recommend the psychological assessment of patients with chronic noncancer pain in order to make informed decisions on the advisability of intervention with opioids [9, 18, 46]. Current instruments used to assess such risk do not

include many of the variables that previous research has found to be associated with opioid misuse and have also been shown to have very limited predictive capacity [10, 47]. In general, there is relatively limited evidence on the accuracy of these tools in predicting misuse [48]. The limitations of these current measures are among the main reasons that Kaye et al., [12] published a review of this area in 2017. They suggested that, at the time of writing, there was still no risk assessment procedure or any empirically determined set of predictive variables that were adequate for identifying patients with chronic noncancer pain at risk of opioid misuse.

In addition, Chou et al., [9] suggested that clinicians should take some basic steps to prevent opioid misuse before starting patients on opioid therapy and addressed the importance of having available a reliable method by which to determine whether patients could take opioids in a responsible and reliable manner. This fundamental issue can only be addressed by the prior identification of subject variables that could predict the risk of misuse.

In summary, there remains an urgent need for comprehensive scientific research using high-quality methodology to identify a set of variables that could reliably predict opioid misuse in chronic pain patients. The results could then be used to guide the development of a practical measure that could be administered to patients by health care providers before making therapeutic decisions.

## Study overview

This project will be led by a research group from the Faculty of Psychology at the University of Málaga (Spain), supported by grants from the Spanish Ministry of Science and Innovation (PID2019-106086RB-I00) and the Regional Government of Andalusia (HUM-566).

Two studies will be conducted, each with a different primary objective. The primary objective of the first study (Study 1) will be to evaluate a theoretical model of opioid drug misuse that includes the key variables that have been identified in previous research as predictors of adherence to appropriate opioid treatment use in patients with chronic noncancer pain who are receiving opioids. The primary objective of Study 2 will be to develop and evaluate the validity of a brief instrument that will include the aspects demonstrated to be most relevant in predicting the misuse of prescription opioids in Study 1.

### Hypothesis

Table 1 shows the hypotheses of the present project.

## Methods

### Participants

The participants will be individuals with chronic noncancer pain attending three Pain Units in the following centres: Hospital Clínico Universitario Virgen de la Victoria (Málaga), Hospital Regional Universitario (Málaga), and Hospital Costa del Sol (Marbella, Málaga). These individuals will either be undergoing pharmacological treatment that includes opioid analgesics (Study 1, N = 400) or are going to commence such treatment (Study 2, N = 250)

Inclusion criteria:

- Chronic noncancer pain.

- Being older than 18 years.

- Ability to understand and sign the informed consent form.

Exclusion criteria:

**Table 1. Hypotheses.**

| | | |
|---|---|---|
| *Study 1*: Testing the hypothetical model | There is a significant positive association between prescription opioid misuse and H1 to H17: | H1. Family history of substance abuse |
| | | H2. Personal history of substance abuse |
| | | H3. Being aged between 16 and 45 years |
| | | H4. History of preadolescent sexual abuse |
| | | H5. Psychological disease (ADD, OCD, bipolar, schizophrenia) |
| | | H6. History or current diagnostic of depression |
| | | H7. A history of smoking and being a current smoker |
| | | H8. Body mass index greater than 30 |
| | | H9. Symptoms of depression |
| | | H10. Symptoms of anxiety |
| | | H11. Symptoms of Post-Traumatic Stress Disorder (PTSD) |
| | | Concerns about medication and medication use: (H12) perceiving medication as needed (H13) concern about negative scrutiny, and (H14) concerns about tolerance |
| | | H15. Pain catastrophizing |
| | | H16. Impulsivity |
| | | H17. Anxiety sensitivity |
| | There is a significant negative association between prescription opioid misuse and H18: | H18. Pain acceptance |
| | There are gender differences in the correlates of risk for opioid misuse (H19 and H20): | H19. Opioid misuse in women will be more strongly associated with emotional issues and affective distress, with opioid medications being used to help stabilize their mood and relax. Therefore, symptoms of anxiety, depression, or PTSD are expected to have higher correlations with opioid misuse in women than in men. |
| | | H20. Men may be more strongly affected by environmental factors involved with associating with others who are prone to using illicit substances. Thus, a history of or current addiction to opioids, alcohol, or other prescription or nonprescription drugs in family or friends are all expected to have higher correlations with prescribed opioid misuse in men than in women. |
| *Study 2*: Development of a measure to assess the risk of prescribed opioid misuse, and the analyses of its reliability, validity, sensitivity, specificity and accuracy. | The total score of the new scale will show (H21-H23): | H21. A moderate correlation with the *Opioid Risk Tool* (ORT-10; Webster and Webster, 2005) (construct validity). In the case that domains included in the ORT are also included in the new scale, the magnitude of the expected correlation will be higher. |
| | | H22. Higher levels of sensitivity, specificity, and accuracy than the ORT. |
| | | H23. High levels of accuracy ($\geq$95% correct classification rate), sensitivity ($\geq$95%), and specificity ($\geq$95%) for classifying patients as being at risk of opioid misuse at 6 months and 12 months after the initial prescription of opioids for chronic pain management. |

- Incapable of understanding the instructions.

- Current (or history of) psychotic symptoms.

- Lack of fluency in spoken and written Spanish.

**Number of participants.** *Study 1. Testing the hypothetical model.* According to Freeman [49], in logistic regression analysis, the sample size must be [n = 10 * (k + 1)] (where k = number of variables). This study will require at least 190 participants [10 * (18+1) = 190]. However, the total number of participants will be 200. Furthermore, to cross-validate the

model in a new sample, a further 200 patients will be needed. Therefore, Study 1 will include a total of 400 patients.

*Study 2*: *Development of a measure to assess the risk of prescribed opioid misuse and the analysis of its reliability and validity*. The aforementioned formula [49] will be used to test the predictive model: however, the number of variables will depend on the results of Study 1. In any case, we will increase the resulting number of participants by 30% in order to safeguard against any dropout at the various assessment stages during the longitudinal study. The aforementioned criteria will be applied [50–52] to test the psychometric properties of the instrument and to determine sample size. However, research has shown that a minimum subject-to-variable ratio of 5-to-1 in linear regression analysis can produce stable and reasonably accurate estimates of the regression coefficients [53]. In summary, in the absence of knowing the number of domains that will be included in the new scale, it seems reasonable to plan for recruiting a sample of 250 patients to take part in the three assessment sessions.

## Procedure

**Study 1: Testing the hypothetical model.**   *Phase 1*. Identify the variables that predict opioid misuse. At the end of their visit to their pain specialist, the patients who fulfil the eligibility criteria will be informed of the study aims and their participation will be requested. Written informed consent will be obtained prior to data collection. Each participant will take part in a semi-structured interview with a psychologist from the research team to obtain demographic, social, or medical history data. A battery of questionnaires will also be completed by each participant in the assessment session. These questionnaires will measure the variables that we expect to be related to opioid misuse and the current misuse of prescription opioids. In order to recruit the necessary number of participants, 140 patients with chronic noncancer pain receiving pharmacological treatment that includes opioid analgesics will be interviewed in each Pain Unit. The patients will always be assessed in their usual health centre.

*Phase 2*. Cross-validation. Based on the results of the regression analysis conducted in Study 1, the variables that predict opioid misuse in a sample of 200 patients will be identified. The model will then be cross-validated with a different sample of 200 patients with the same characteristics.

**Study 2: Development of a measure to assess the risk of prescribed opioid misuse and the analysis of its reliability and validity.**   Study 2 will include 3 phases: development of the initial item pool (Phase 1) and a prospective analysis with three measurement stages (Phases 1, 2 and 3).

*Phase 1*. Development of the initial item pool. Once the variables that predict opioid misuse have been identified, we will develop a scale to assess these domains Firstly, the research team will choose an initial pool of 6 items per domain and a panel of experts in methodology will rate the items in relation to the appropriateness of their formulation, with changes being made following their suggestions. Secondly, a brief and precise definition of each construct will be developed. Thirdly, an expert panel will be carefully chosen to ensure their knowledge of the content domain. The experts will be asked to rate the representativeness and relevance of each item in relation to the definition of the constructs. Inter-rater agreement will be calculated, discrepancies will be discussed, and the process will be repeated until acceptable levels of agreement are achieved (.70 to .80) [54, 55]. The three items most representative of each domain will be chosen to be part of the initial instrument. A pilot study with a group of at least 10 patients will be conducted to test the readability and comprehensibility of the items and changes will be made if necessary.

At the end of their visit to their pain specialist, the patients who fulfil the eligibility criteria (i.e., those who are going to begin treatment with opioid analgesics) will be informed of the

study aims and their participation will be requested. Written informed consent will be obtained prior to data collection. Each participant will take part in a semi-structured interview with a psychologist from the research team to obtain demographic, social, or medical history data. The first 10 patients will be asked to participate in the pilot study of the instrument.

Subsequent patients (a total of 250) will then complete the revised scale (EDRAO) and the Opioid Risk Tool (ORT-10) in the first assessment session. Patients will be informed of the longitudinal design of the study.

*Phases 2 and 3*. Prospective analysis. Second and third assessment sessions. After the initial session, the patients will be contacted twice at 6-month intervals (for each of two check-up visits to the pain specialist) to make an appointment for further evaluation.

In the second assessment session (6 months after the initial visit) and the third (12 months after the initial visit), patients will again complete the EDRAO for test-retest purposes and the COMM to measure current misuse of opioids. In addition, patients will be monitored for indexes of misuse. A pain specialist (i.e., member of the working team) will provide us with information about indexes 1–2 and the psychologist will ask patients about indexes 3–11 in a semi-structured interview (see list of misuse indexes below, in *Variables and Instruments*). Table 2 shows the procedure.

## Variables and instruments

### Study 1: Testing the hypothetical model.

a. *Predictors of opioid misuse*. Each participant will take part in a semi-structured interview with a psychologist to measure the following variables:

- <u>Demographic, social, and medical history data</u>. We will record the medication that patients are currently taking and the medication prescribed by the specialist (including doses) in their first visit in Pain Unit.

- <u>Pain intensity.</u> Patients will be asked to rate their mildest, average, and worst pain during the past 2 weeks, as well as their current pain, on a scale ranging from 0 to 10, with a "0" indicating "no pain" and "10" indicating pain as "intense as you could imagine". A composite pain intensity score will be calculated for each subject by calculating the average of the mildest, average, worst, and current pain. Composites of 0-to-10 ratings are highly reliable measures of pain intensity in chronic pain patients [56].

In order to measure the variables included in Hypothesis H1-H6, the Spanish version of the *Opioid Risk Tool (ORT-10)*. [22] will be used. The Spanish version is currently being validated as part of a PhD dissertation by our research team. The ORT-10 is a 10-item scale that assesses five domains or risk factors for prescription opioid misuse derived from clinical experience and a literature review conducted by the developers of the measuring instrument. The domains include a family and personal history of substance abuse, age, a history of preadolescent sexual abuse, and specific psychological disorders (i.e., having a diagnosis of attention deficit disorder, obsessive-compulsive disorder, bipolar disorder, schizophrenia, or depression). Each risk factor is weighted and attributed a point value believed to reflect its risk relative to the other risk factors. Items are scored and the total score can range from 0 to 26. A score of 3 or less indicates a low risk of future opioid misuse, a score of 4 to 7 indicates a moderate risk of opioid misuse, and a score of 8 or higher indicates a high risk of opioid misuse.

All the other predictor variables will be measured (also during the interview) with the following instruments:

**Table 2. Procedure.**

| Study: Objective | Phases: Objective | Sample | Variables |
|---|---|---|---|
| *Study 1*: Testing the hypothetical model (N = 400) | *Phase 1*: Identify the variables that predict opioid misuse | Individuals who undergo pharmacological treatment that includes opioid analgesics (N = 200) | • Demographic, social, and medical history data |
| | | | • Pain intensity |
| | | | • Family and personal history of substance abuse |
| | | | • History of preadolescent sexual abuse |
| | | | • Specific psychological disorders (attention deficit disorder, obsessive-compulsive disorder, bipolar disorder, schizophrenia or depression) |
| | | | • Anxiety and depression symptoms |
| | | | • Post-Traumatic Stress Disorder |
| | | | • Concerns about medication and medication use |
| | | | • Pain catastrophizing |
| | | | • Impulsiveness |
| | | | • Anxiety sensitivity |
| | | | • Pain acceptance |
| | | | • Index of opioid misuse |
| | *Phase 2*: Cross-validation of model | The same as phase 1 (N = 200) | Based on the results of the regression analysis conducted in phase 1, the variables that predict opioid misuse |
| *Study 2*. Development of a new measure to assess the risk of prescribed opioid misuse and the analyses of its reliability and validity (N = 250) | *Phase 1*: Development of the initial item pool | An expert panel | Selection of three items most representative of each domain will be chosen to be part of the initial instrument |
| | | Pilot group to test the readability and comprehensibility of the items (N = 10) | • The new instrument (EDRAO) |
| | | Patients who are going to start pharmacological treatment that includes opioids (N = 250) | • EDRAO |
| | | | • A standardized instrument to measure the risk of opioid misuse (ORT-10) |
| | *Phase 2*: Prospective analysis (2nd assessment) | The sample in phase 2, 6 months later (N = 250) | Indexes of misuse |
| | | | Indexes of misuse (COMM) |
| | *Phase 3*: Prospective analysis (3rd assessment) | The sample in phase 3, 6 months later (N = 250) | Indexes of misuse |
| | | | Indexes of misuse (COMM) |

Symptoms of Depression and/or Anxiety. *Hospital Anxiety and Depression Scale (HADS)* [57]. This is a self-report scale that contains two 7-item scales, one for anxiety and one for depression, each of which are scored on a 4-point scale. A score from 8 to 10 indicates possible cases and a score of 10 or more indicates probable cases. The Spanish version of the scale shows appropriate reliability and validity [58].

The presence of symptoms of Post-Traumatic Stress Disorder (PTSD). The Spanish version of the Posttraumatic Stress Disorder Checklist-Civilian Version (PCL-5) [59] will be used to assess PTSD as described in DSM-5. This instrument comprises a 20-item self-report checklist which is rated on a 5-point Likert-scale (ranging from 1 = never to 5 = very often) to indicate the degree to which each particular symptom had been experienced by the participant over the past month.

Concerns about medication and medication use. Participants' concerns about pain medication will be measured using the short form of the *Pain Medication Attitude Questionnaire- 14 (PMAQ-14)* [26]. The Spanish version of this scale is currently being adapted by our research group as part of a PhD dissertation. It comprises 14 items grouped into 7 domains (addiction, need, scrutiny, side effects, tolerance, mistrust, and withdrawal). Each item is scored on a

6-point Likert-scale ranging from 0 ("never true") to 5 ("always true"). The scale shows good reliability and validity [26].

Pain catastrophizing. *Pain Catastrophizing Scale (PCS)*. This scale is a 13-item measure in which respondents indicate on a 5-point scale the degree to which they experience various thoughts and feelings while in pain [60]. It consists of three subscales assessing rumination, magnification, and helplessness and also provides a total score on catastrophizing. The total score alone will be used in this study. The Spanish version of the scale shows good reliability and validity and its internal consistency is high [61].

Impulsiveness. The Spanish version of the Barratt Impulsiveness Scale for adults (BIS-11) [62–64] will be used to measure impulsiveness. It is composed of 30 items that are answered on a 4-point scale (from 1 = rarely or never to 4 = always) where 4 indicates the most impulsive response. The higher total score, the higher level of impulsiveness. The Spanish Version of the scale shows good reliability and validity [64].

Anxiety sensitivity. *Anxiety Sensitivity Index-3 (ASI-3)*. This instrument is an 18-item questionnaire where respondents indicate the degree to which they fear the negative consequences of anxiety symptoms on a 5-point Likert-type scale [65]. The results of validation studies provide cross-cultural evidence for construct validity and the concurrent validity of the Spanish ASI-3 [66].

Pain acceptance. *Chronic Pain Acceptance Questionnaire (CPAQ)*. We will apply the Spanish version of the questionnaire (CPAQ-SV) [67, 68]. This instrument consists of 20 items. Similar to the original questionnaire, the CPAQ-SV yields a total score and two subscale scores for Pain Willingness and Activity Engagement. The subscales of the CPAQ-SV show good internal consistency [68]. Two studies on the CPAQ-SV [68, 69] have supported the validity of a 20-item version with two subscales corresponding to two independent factors. In addition, the CPAQ-SV demonstrates good criterion validity.

b. *Indexes of opioid misuse*. Current misuse of prescribed opioids. The Spanish translation of the *Current Opioid Misuse Measure (COMM)* [70] will be used. The Spanish version of this scale is currently being adapted by our research group as part of a PhD dissertation. It is a brief self-assessment instrument to monitor chronic pain patients on opioid therapy. The COMM contains 17 items rated from 0 = "never" to 4 = "very often" (see Table 1). The COMM was developed to track patient status over time, and so the items included in the COMM can be used repeatedly and provide an estimate of the patients' "current" status. Thus, the items cover a 30-day period (i.e., "In the past 30 days. . .") and only behaviour that could change from time to time are included (i.e., historical items are excluded). The 17 items are summed to create a total score. A total score of 9 or more is considered a positive indication of opioid misuse. Meltzer et al., [71] found higher COMM scores in patients with chronic pain who had a prescription drug use disorder than in those who did not have the disorder, thus supporting the construct validity of the instrument.

We will measure opioid misuse from two perspectives; self-reports via structured interviews and physician reports. Thus, the following behaviours are some indexes of misuse that will be measured:

1. Requesting more opioid medication before the next appointment with the physician

2. Going to primary care or emergency services in order to obtain more opioid medication before the next appointment

3. Requesting more opioids than prescribed by the doctor in the pharmacy

4.  Requesting more opioids via a third person or by Internet

5.  Admitting to seeking euphoria from opioids

6.  Admitting to wanting opioids for anxiety

7.  Requesting opioids from other providers

8.  Unauthorized dose escalation

9.  Reporting lost or stolen prescriptions

10. Requesting refills instead of a visit to the clinic

Behaviours 1 to 2 will be identified through the patients' DIRAYA cards, which are their electronic health records (Historia Digital de Salud del Ciudadano) held in the Andalusian Public Health System. They can be readily accessed by the pain physicians (i.e., the four components of the working team). Behaviours 3 to 10 could be detected through interviews with the patients. Patients with more than 2 indexes of misuse will be considered positive for prescribed opioid misuse. Therefore, a total score of 9 or higher on COMM and/or more than 2 indexes of misuse will be considered positive for prescribed opioid misuse. This will be a cheap, easy, and objective method to evaluate the patients' misuse of prescribed opioid medicines. It is worth mentioning that this study excludes urine toxicology screens as an appropriate measure of misuse. Since the sample will be composed of patients undergoing an opioid treatment regimen, the presence of these substances would be expected. This screen tests for the presence of an additional non-prescribed drug in urine [12], which is not our aim. In any case, urine and blood opioid tests are not currently used in the Andalusian Public Health System. In fact, Fentanyl misuse is currently detected through the DIRAYA card in three of the participant pain units. Moreover, urine or blood analysis would be of little value for detecting quick-release transdermal Fentanyl treatment, and therefore such tests would not be appropriate given the aims of this project.

**Study 2: Develop a measure to assess the risk of prescribed opioid misuse and examine its reliability and validity.**   The new scale (EDRAO) will be applied. In order to test its validity, the *Opioid Risk Tool* (ORT-10 [22]; see above) and all of the indexes of misuse measured in Study 1 (explained above) will be used at the two assessment points of Study 2.

## Statistical analyses

Data analyses will be conducted using the Statistical Package for the Social Sciences version 26.0 (SPSS; Chicago, IL, USA).

**Study 1: Testing the hypothetical model.**   Firstly, socio-demographic data will be expressed as frequencies and percentages and continuous data will be expressed as means and standard deviations. Since opioid misuse is a dichotomous variable, logistic regression analysis will be used to identify significant associations between the set of variables included in the hypothetical model and opioid misuse. The patients' time in pharmacological treatment with opioids, pain intensity, intake of benzodiazepines, type of chronic pain, and sex will be included as covariates in the regression analyses. The following steps will be taken to fit a parsimonious model with strong predictive covariates only. The potential risk factors that have been identified in the literature will be evaluated for potential univariate associations with opioid misuse. The independent-sample t-test will be used to analyse continuous variables and $\chi^2$ or Fisher's exact tests will be used to analyse categorical or discrete variables. Multivariable conditional logistic regression will be used to identify the combined effect of several risk factors. Those with a univariate p value $\leq$.10 will be considered for inclusion in the model. Using

a forward stepwise procedure, the variables that achieved a p value ≤.05 will remain in the final model. The results will be expressed as odds ratios (ORs) with 95% confidence intervals (CIs) and p values. A two-tailed P value of < .05 will be used as a cutoff for statistical significance.

**Study 2: Developing a measure to assess the risk of prescribed opioid misuse and to examine its reliability and validity.** As mentioned, an instrument will be developed to measure the constructs that, according to the results of Study 1, were significantly associated with opioid misuse. Since our aim is to develop a brief instrument, 3 items per construct will be selected by experts to assess each variable.

**Item analyses.** Firstly, in order to construct the shortest instrument possible, reduce the number of items, and retain only those with the highest discriminative power, t-tests will be run to determine which of the selected items identified patient status at the follow-ups (opioid misuse vs opioid non-misuse). Effect sizes (Cohen's D) will be calculated and an effect size of 0.40 will be used as a cutoff to select items.

**Sensitivity, specificity, and accuracy.** Secondly, we will test the sensitivity, specificity, and accuracy of the final set of items. Receiver operating characteristic (ROC) curve analysis will be used to test the predictive validity of the final items of the EDRAO at follow-ups 1 and 2 as well as to determine the cut-off scores of the newly developed instrument. To determine the extent to which the values are influenced by outliers, the sample will be repeatedly randomly divided in half and the ROC curve will be calculated for each half separately.

**Scale analyses.** The stability of the instrument will be tested in a subsample of the participants and factor analysis will be used to study its dimensionality. In order to test the construct validity of EDRAO, we will analyse correlations between the scale and ORT-10.

## National and local approvals

The project will be conducted in accordance with the Declaration of Helsinki and has received ethical clearance by the Institutional Ethics Review Board (ERC UMA) and the Regional Hospital Ethics Committee. This study is supported by grants from the Spanish Ministry of Science and Innovation (PID2019-106086RB-I00) and the Regional Government of Andalusia (HUM-566, P07-SEJ-3067).

## Discussion

Several studies have highlighted the adverse effects of opioid medication due to their misuse [9, 11, 12, 22, 23]. In this regard, the increased use of opioids for long-term treatment increases the risk of death due to both unintentional overdose and cardiorespiratory problems. Therefore, prior risk assessment has emerged as a crucial step in making informed decisions on the risks of starting treatment with opioid drugs. Several treatment guidelines have strongly recommended assessing the risks and benefits of pharmacological treatment with opioids [9]. They have also recommended the psychological assessment of patients with chronic noncancer pain in order to make informed decisions on the advisability of interventions with opioids [18, 46]. The most commonly used measures for predicting the potential for opioid misuse do not include many of the variables shown to be associated with opioid use in previous research and have been shown to have very limited predictive capacity. Furthermore, these measurement instruments have not been adapted to the Spanish population and are therefore not being used in Spanish health centres. It is therefore essential to have a reliable method that can determine which patients could take opioids in a responsible and reliable manner. This fundamental issue can only be addressed by the prior identification of subject variables that could predict the risk of misuse. In summary, there remains an urgent need for comprehensive scientific

research using high-quality methodology to identify a set of variables that would reliably predict opioid misuse in chronic pain patients. The results could then be used to develop a practical measure that could be administered to patients by health care providers before making therapeutic decisions. This project will fill this gap by providing a comprehensive model to identify the factors most closely associated with subsequent opioid misuse and a reliable, valid, and useful instrument that would enable health care providers to make decisions on opioid prescription and on the required level of monitoring and follow-up.

In addition, our group is currently adapting the Spanish versions of three valuable measurement instruments as part of a PhD dissertation, and we intend to make use of these in this project. Increasing the visibility of these instruments, alongside the development of a reliable and valid scale for assessing the risk of opioid misuse, will have a very positive impact on Pain Units and provide new tools to be used in clinical settings. The collaboration of three pain units and three pain specialist physicians will ensure the impact of the results in such health centres. Furthermore, the dissemination of our findings at the *Andalusian Pain Society* and *Spanish Pain Society* conferences will maximize the national impact of the project.

Finally, the results of the project will help to alleviate the serious consequences of the increasing use of opioid medications through the prior identification of subject variables that could predict the risk of prescription opioid misuse in patients with chronic noncancer pain. This will lead to very relevant changes in Pain Units by having a positive impact on the wellbeing of these patients.

This project fulfils the objectives of the Spanish Strategy of Science and Technology and Innovation (2017–2020) and, more specifically, those of Challenge 1 "Health, demographic change, and well-being".

## Conclusions

The results of this project will provide medical professionals with an instrument to detect the risk of opioid misuse in patients with chronic noncancer pain. The accurate and early detection and classification of patients who are at the greatest risk of opioid misuse will allow alternative medications to be considered. Should physicians decide that treatment with opioids is indicated, it will also help to determine the level of monitoring that will be needed based on the risk group to which the patient has been classified.

## Acknowledgments

We thank Elena R. Serrano-Ibánez, Gema T. Ruiz-Párraga, Rocío de la Vega, and Mark P. Jensen for contributing to the review of the project. We also express our gratitude to Sergio Escorial for reviewing the statistical analysis and to Mariano Fernández and José Manuel González-Mesa who reviewed the procedure.

## Author Contributions

**Conceptualization:** Carmen Ramírez-Maestre, Alicia E. López-Martínez, Rosa Esteve.

**Methodology:** Carmen Ramírez-Maestre, Alicia E. López-Martínez, Rosa Esteve.

**Project administration:** Carmen Ramírez-Maestre, Alicia E. López-Martínez.

**Supervision:** Carmen Ramírez-Maestre, Alicia E. López-Martínez, Rosa Esteve.

**Writing – original draft:** Carmen Ramírez-Maestre.

**Writing – review & editing:** Carmen Ramírez-Maestre, Alicia E. López-Martínez, Rosa Esteve.

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
