## [Decision Letter · Decision Letter 0]

5 Mar 2021

PONE-D-21-02639

Predictive variables of prescription opioid misuse in patients with chronic non-oncologic pain. Development of a risk detection scale : A registered report protocol

PLOS ONE

Dear Dr. Ramírez-Maestre,

Thank you for submitting your manuscript to PLOS ONE. After careful consideration, we feel that it has merit but does not fully meet PLOS ONE’s publication criteria as it currently stands. Therefore, we invite you to submit a revised version of the manuscript that addresses the points raised during the review process.

We look forward to receiving your revised manuscript.

Kind regards,

Vijayaprakash Suppiah, PhD

Academic Editor

PLOS ONE

Journal Requirements:

2. We recognize that, as this is a Registered Report Protocol, you do not yet have data available to share. However, as per PLOS' guidelines, we do expect authors to provide a Data Availability Statement and outline your data management plan and/or state how you expect their data to be made available (https://journals.plos.org/plosone/s/criteria-for-publication#loc-7). To this effect, please confirm that you will be able to make your data available when your study is completed and if it is accepted for publication.

4. Please ensure that you include a title page within your main document. You should list all authors and all affiliations as per our author instructions and clearly indicate the corresponding author.

5. Please include your tables as part of your main manuscript and remove the individual files. Please note that supplementary tables (should remain/ be uploaded) as separate "supporting information" files

Reviewers' comments:

Reviewer's Responses to Questions

**Comments to the Author**

1. Does the manuscript provide a valid rationale for the proposed study, with clearly identified and justified research questions?

Reviewer #1: Yes

Reviewer #2: Yes

2. Is the protocol technically sound and planned in a manner that will lead to a meaningful outcome and allow testing the stated hypotheses?

Reviewer #1: Yes

Reviewer #2: Yes

3. Is the methodology feasible and described in sufficient detail to allow the work to be replicable?

Reviewer #1: Yes

Reviewer #2: No

4. Have the authors described where all data underlying the findings will be made available when the study is complete?

Reviewer #1: Yes

Reviewer #2: No

5. Is the manuscript presented in an intelligible fashion and written in standard English?

Reviewer #1: Yes

Reviewer #2: No

6. Review Comments to the Author

You may also provide optional suggestions and comments to authors that they might find helpful in planning their study.

Reviewer #1: Major comments

The authors are planning to conduct two studies, each with a different primary objective: (1) conducting a comprehensive study to identify the factors most closely associated with subsequent opioid misuse and, (2) developing a brief, reliable, valid, and useful instrument informed by this research that would enable healthcare providers to make decisions regarding opioid prescription and the needed level of monitoring and follow-up.

In study 1, A battery of questionnaires will also be completed by each participant in the assessment session. These questionnaires will measure the variables that we expect to be related to opioid misuse, and the current misuse of prescription opioids. And then, on the basis of the results of the regression analysis conducted in Study 1, those variables that predict opioid misuse in a sample of 200 patients will be identified. A cross validation of the model will then be conducted with a different sample of 200 patients with the same characteristics.

Subsequently, in study 2, to develop a measure to assess risk of prescribed opioid misuse, and to analyze its reliability and validity, a longitudinal design with three assessment sessions will be applied.

This referee think that this study protocol is well designed and the manuscript is well written, but there are several grammatical errors on the manuscript.

Reviewer #2: The authors have presented a nice presentation of two protocols that will be conducted in order to identify some predictors of opioid misuse and then to develop a scale to be used in clinical practice

The topic is surely important and highly debated and the protocols try to add some important informations to scientific debate

Unfortunately I think that there are some major concerns that should be addressed in order to improve the quality of the manuscript, of the protocols and in order to have a better reproducible dataset.

The first major concern that should be addressed is about the concept of "misuse" defined in the title. Opioids in chronic pain patients can lead not only to misuse but also to abuse. Please discuss better this two problems and define better how authors' protocols can help clinicians to identify the two different problems

The second major concern is about the definition of chronic pain. How do the authors define "chronic pain"?Please provide a definition : temporal definition or pathophysiological definition? It is strange that authors have considered several psychological and demographic factors to be correlated to misuse/abuse but they have not considered if different chronic pain syndromes could be related to different risk for misuse/abuse. Please discuss it and provide explanations (better if you can consider them)

The third concern is about the sample size explanation: the authors stated "the sample size must be: [n = 10 * (k + 1)] (being k= number of variables). In this study we will need at least 180 participants [10 * (16+1) = 160]." How do you decide that K is 16?Please discuss it more in detail

Finally minor concerns

You have stated that there will be limitations to the available data: please provide an explanation

It should be better to revise all the text with a native English speaker

7. PLOS authors have the option to publish the peer review history of their article (what does this mean?). If published, this will include your full peer review and any attached files.

Reviewer #1: **Yes: **Young-Chang Arai

Reviewer #2: No

---

## [Author Response · Author response to Decision Letter 0]

3 Apr 2021

Dear Dr. Vijayaprakash Suppiah,

We very much appreciate your new suggestions regarding our paper (PONE-D-21-02639

Predictive variables of prescription opioid misuse in patients with chronic noncancer pain. Development of a risk detection scale: A registered report protocol). We have revised the paper accordingly and hope the new version is now acceptable for publication.

This is a Registered Report Protocol and so we do not yet have data available to share (not even a minimal data set). However, I confirm that we will be able to make our data available when our study is completed and if it is accepted for publication.

In the following, we explain how the reviewers' suggestions have been taken into account in the new version. Changes in the text can be found in red.

Journal’s requirements:

Please ensure that you include a title page within your main document. You should list all authors and all affiliations as per our author instructions and clearly indicate the corresponding author

Answer. The title page has been included within our main document

Please include your tables as part of your main manuscript and remove the individual files. Please note that supplementary tables (should remain/ be uploaded) as separate "supporting information" files.

Answer. Following the Journal´s requirements, the tables have been included as part of our main manuscript.

Reviewer comments:

Reviewer #1 (R#1)

This referee think that this study protocol is well designed and the manuscript is well written, but there are several grammatical errors on the manuscript.

Thank you very much for your positive comments. A professional translator (native English speaker) has reviewed again the text.

Reviewer #2 (R#2)

1. The first major concern that should be addressed is about the concept of "misuse" defined in the title. Opioids in chronic pain patients can lead not only to misuse but also to abuse. Please discuss better this two problems and define better how authors' protocols can help clinicians to identify the two different problems

Answer. We agree with reviewer. In fact, we just wanted to assess the overuse of opioids prescribed. New text has been included in the introduction section (new text shown in red), explaining that the misuse of opioids "... has been defined as the use of opioids in a manner other than how they are prescribed (Martel, Edwards, & Jamison, 2020). Prescription opioid misuse may involve a wide range of behaviours, including overuse (Martel et al., 2020). In our project, misuse is defined as the overuse of prescribed opioids (i.e., taking more opioids than prescribed)” (lines 79-82)

New reference:

14- Martel MO, Edwards RR, Jamison RN. The relative contribution of pain and psychological factors to opioid misuse: A 6-month observational study. American Psychologist. 2020; 75(6): 772. https://psycnet.apa.org/doi/10.1037/amp0000632

2. The second major concern is about the definition of chronic pain. How do the authors define "chronic pain"? Please provide a definition : temporal definition or pathophysiological definition? It is strange that authors have considered several psychological and demographic factors to be correlated to misuse/abuse but they have not considered if different chronic pain syndromes could be related to different risk for misuse/abuse. Please discuss it and provide explanations (better if you can consider them)

Answer. Thank you for your comment. We have used a temporal definition, according to the proposal of Treede et al., (2019). Following R#2 suggestion, the following sentence has been included in the introduction section: 

“Chronic pain is pain that lasts or recurs for longer than 3 months (Treede et al., 2019)”. (line 70-71)

We found your point about considering if different chronic pain syndromes could be related to different risk for misuse/abuse to be very interesting. However, including more variables in this project would increase the difficulty of recruiting the sample because of having to increase the required number of participants. It is relevant to bear in mind that this project has received funding for only 4 years, and this period might not be enough if the complexity of the study is increased. 

In any case, according to R#2´s suggestion, the type of chronic pain (following the IASP new classification 2019) will be included in the analysis as covariables:

“The patients’ time in pharmacological treatment with opioids, pain intensity, intake of benzodiazepines, type of chronic pain, and sex will be included as covariates in the regression analyses”. (line 377)

3. The third concern is about the sample size explanation: the authors stated "the sample size must be: [n = 10 * (k + 1)] (being k= number of variables). In this study we will need at least 180 participants [10 * (16+1) = 160]." How do you decide that K is 16? Please discuss it more in detail

Answer: We absolutely agree with the reviewer. This is a mistake. Eighteen variables will be included in the regression model (Study 1) (see Table 2):

• Demographic, social, and medical history data:

o Sex, age, time in pharmacological treatment with opioids, intake of benzodiazepines, and type of chronic pain (5 co-variables). 

• Pain intensity

• Family and personal history of substance abuse

• History of preadolescent sexual abuse

• Specific psychological disorders (attention deficit disorder, obsessive-compulsive disorder, bipolar disorder, schizophrenia or depression).

• Anxiety and depression symptoms (2 variables)

• Post-Traumatic Stress Disorder

• Concerns about medication and medication use

• Pain catastrophizing

• Impulsiveness

• Anxiety sensitivity

• Pain acceptance

• Index of opioid misuse

Therefore, that text has changed:

“According to Freeman (49), in logistic regression analysis, the sample size must be [n = 10 * (k + 1)] (where k = number of variables). This study will require at least 190 participants [10 * (18+1) = 190]." 

4. You have stated that there will be limitations to the available data: please provide an explanation.

Answer: I apologise. What we wanted to say is that as a Registered Report Protocol, we do not yet have data available to share (not even a minimal data set).

5. It should be better to revise all the text with a native English speaker

Answer: Following R#2’s suggestion, a professional translator (native English speaker) has reviewed the text again.

---

## [Decision Letter · Decision Letter 1]

29 Apr 2021

Predictive variables of prescription opioid misuse in patients with chronic noncancer pain. Development of a risk detection scale : A registered report protocol

PONE-D-21-02639R1

Dear Dr. Ramírez-Maestre,

We’re pleased to inform you that your manuscript has been judged scientifically suitable for publication and will be formally accepted for publication once it meets all outstanding technical requirements.

Kind regards,

Vijayaprakash Suppiah, PhD

Academic Editor

PLOS ONE

Reviewers' comments:

Reviewer's Responses to Questions

**Comments to the Author**

1. Does the manuscript provide a valid rationale for the proposed study, with clearly identified and justified research questions?

Reviewer #1: Yes

Reviewer #2: Yes

2. Is the protocol technically sound and planned in a manner that will lead to a meaningful outcome and allow testing the stated hypotheses?

Reviewer #1: Yes

Reviewer #2: Yes

3. Is the methodology feasible and described in sufficient detail to allow the work to be replicable?

Reviewer #1: Yes

Reviewer #2: Yes

4. Have the authors described where all data underlying the findings will be made available when the study is complete?

Reviewer #1: Yes

Reviewer #2: Yes

5. Is the manuscript presented in an intelligible fashion and written in standard English?

Reviewer #1: Yes

Reviewer #2: Yes

6. Review Comments to the Author

You may also provide optional suggestions and comments to authors that they might find helpful in planning their study.

Reviewer #1: After this referee read this version and found out that the authors amended the manuscript well, the referee is satisfied with the content.

Reviewer #2: The authors have deeply revised the text accordingly reviewers' suggestions improving the quality of the data presented

7. PLOS authors have the option to publish the peer review history of their article (what does this mean?). If published, this will include your full peer review and any attached files.

Reviewer #1: **Yes: **Young-Chang Arai

Reviewer #2: No

---

## [Editor Report · Acceptance letter]

3 May 2021

PONE-D-21-02639R1 

Predictive variables of prescription opioid misuse in patients with chronic noncancer pain. Development of a risk detection scale: A registered report protocol 

Dear Dr. Ramírez-Maestre:

I'm pleased to inform you that your manuscript has been deemed suitable for publication in PLOS ONE. Congratulations! Your manuscript is now with our production department. 

Kind regards, 

on behalf of

Dr. Vijayaprakash Suppiah 

Academic Editor

PLOS ONE